



# Seismo-acoustic energy partitioning of a powder snow avalanche

Emanuele Marchetti[1], Alec van Herwijnen[2], Marc Christen[2], Maria Cristina Silengo[1], and
Giulia Barfucci[1]

[1]Department of Earth Sciences, University of Firenze, Firenze, Italy.
[2]WSL Institute for Snow and Avalanche Research SLF, Davos, Switzerland.

**Correspondence:** Emanuele Marchetti (emanuele.marchetti@unifi.it)

**Abstract.** While flowing downhill, a snow avalanche radiates seismic waves in the ground and infrasonic waves in the atmosphere. Seismic energy is radiated by the dense basal layer flowing above the ground, while infrasound energy is likely radiated by the powder front. However, the mutual energy partitioning is not fully understood. We present infrasonic and seismic array data of a powder snow avalanche, that released on 5 February 2016, in the Dischma valley above Davos, Switzerland. A five
element infrasound array and a seven element seismic array were deployed at short distance ($< 500$ m) from each other, and close ($< 1500$ m) to the avalanche path. The avalanche dynamics was modeled by using RAMMS, and characterized in terms of front velocity and flow height. The use of arrays rather than single sensors, allowed us to increase the signal-to-noise ratio, and to identify the event in terms of back-azimuth and apparent velocity of the recorded wave-fields. Wave parameters, derived from array processing, were used to identify the avalanche path and highlight the areas, along the path, where seismic and
infrasound energy radiation occurred. The analysis showed that seismic energy is radiated all along the avalanche path, from the initiation to the deposition area, while infrasound is radiated only from a limited sector, where the flow is accelerated and the powder cloud develops. Recorded seismic signal is characterized by scattered back-azimuth, suggesting that seismic energy is likely radiated by multiple sources acting at once. On the contrary, infrasound signal is characterized by a clear variation of back-azimuth and apparent velocity. This indicates that infrasound energy radiation is dominated by a moving point source,
likely consistent with the powder cloud. Thanks to such clear wave parameters, infrasound revealed particularly efficient for avalanche detection and path identification. While the infrasound apparent velocity decreases as the flow moves downhill, the seismic apparent velocity is quite scattered, but it lowers down to sound velocity during the phase of maximum infrasound radiation. This indicates an efficient process of infrasound to seismic energy transition, that, in our case, increases $\approx 20$ % the recorded seismic amplitude. Such an effect can be accounted for when avalanche magnitude is estimated from seismic
amplitude. Presented results clearly indicate how the process of seismo-acoustic energy radiation by a powder avalanche is very complex, and likely controlled by the powder cloud formation and dynamics, and is hence affected by the path geometry and snow characteristics.



## 1 Introduction

As a first approximation, powder snow avalanches (PSA) can be described with a two layer model consisting of a dense basal snow layer, with densities of 100-400 kg/m$^3$, and a powder part that develops at the head of the flow, with density of 3-30 kg/m$^3$ (*Issler*, 2003). *Carrol et al.* (2013) provided a detailed analytical description of the powder front of a PSA in terms of an eruption current. They showed how the front evolution is mostly controlled by the amount of snow scoured from the snowpack, as the front moves downhill. This erosion depends on the characteristics of the snowpack, such as density and temperature,

and channel morphology (see e.g. *Louge et al.*, 2012; *Carrol et al.*, 2013). The formation of the powder front is enhanced by the narrowing of the avalanche channel, while the front spreading causes deceleration and consequent collapse of the front. Moreover, *Steinkogler et al.* (2014) showed how the evolution of the powder cloud is affected by the temperature of the snow cover. They showed that the temperature of - 2 degrees C of the scoured snow, is a threshold value between different dynamics, with the cloud formation inhibited for "warm avalanches".

While flowing downhill, the interaction of the dense basal flow with the ground radiates seismic energy (*Sabot et al.*, 1998). Infrasound energy is radiated by the compression of the atmosphere produced mostly by the powder front (see e.g. *Schaerer and Salway*, 1980; *Bedard*, 1989). The ratio between the dense and powder part of a snow avalanche, and hence between the seismic and infrasound energy radiation, is not constant while it depends on the front evolution through time (*Carrol et al.*, 2013).

Seismic measurements have been widely applied to investigate avalanche dynamics and characteristics. *Sabot et al.* (1998) showed that slope changes, and the presence of obstacles on the flow path strongly affect the radiation of seismic energy. Moreover, characteristics of recorded seismic signals depend on snow density and avalanche type and size (*Biescas et al.*, 2003; *van Herwijnen and Schweizer*, 2011b; *Vilajosana et al.*, 2007b). Seismic monitoring techniques deploying multiple sensors along the avalanche path (*Biescas et al.*, 2003; *Vilajosana et al.*, 2007a) or arrays (*van Herwijnen and Schweizer*,

2011a; *Lacroix et al.*, 2012; *Heck et al.*, 2017), that allow to identify the avalanche occurrence within a source-to-receiver distance up to ≈ 3 km. *Hammer et al.* (2017) recorded very large avalanches up to 30 km away.

After the pioneer study by *Bedard* (1989), the use of the infrasound in avalanche monitoring and research has increased significantly (*Chritin et al.*, 1996; *Adam et al.*, 1998; *Comey and Mendenhall*, 2004). *Naugolnykh and Bedard* (1990) suggested that infrasound is possibly induced by the non-stationary motion and/or by the turbulence of the flow. Moreover, they suggested

that the amplitude and frequency of the recorded infrasound signals should scale with the avalanche size and velocity.

Since then, infrasound avalanche observations improved substantially, both in number and accuracy. The development and use of infrasound arrays instead of single sensors (*Scott et al.*, 2007; *Ulivieri et al.*, 2011; *Havens et al.*, 2014; *Marchetti et al.*, 2015), allowed to increase the signal-to-noise ratio and to improve the investigation of the avalanche infrasound signature. Specific wave parameters (back-azimuth and apparent velocity) of recorded signals were used to define array processing pro-

cedures, able to detect medium size snow avalanches at distances of a few kilometers (*Ulivieri et al.*, 2011; *Marchetti et al.*, 2015; *Mayer et al.*, 2018). Moreover, infrasound array derived information were used to remotely evaluate the avalanche front position and velocity through time (*Marchetti et al.*, 2015; *Havens et al.*, 2014).



While geophysical observations of snow avalanches significantly improved the monitoring techniques, a robust conceptual model of seismic and infrasound energy radiation is still missing. Previous studies based on infrasound and seismic records allowed to investigate to some extent the mutual characteristics (*Surinach et al.*, 2001; *Kogelnig et al.*, 2011). However, many open questions remain and using infrasound and seismic signals to infer avalanche size is still debated. It is well known that seismic and infrasound energy interact at the earth free surface and are transmitted between the atmosphere and solid earth (*Ichihara et al.*, 2012). The transmission affects the amplitude of recorded signal and should be considered when signal characteristics are used to constrain the source process or to calculate the energy of the event.

In this work we present a combined seismic and infrasound array analysis for a snow slab avalanche that occurred on 5 February 2016, in the Dischma Valley, south of Davos, Switzerland. The event was recorded by a seismic and an infrasound array located nearby (less than 1500 m) the path. The data obtained from the seismic and the infrasound array are used to investigate the mutual energy radiation as a function of the front position along the avalanche path. To investigate the properties of recorded signals as a function of event characteristics, the avalanche was modeled using the avalanche simulation software RAMMS (*Christen et al.*, 2010) .

## 2 Study site

During the winter of 2015-2016 a seismic array and an infrasound array were colocated in the Dischma valley, south of Davos, in the Eastern Swiss Alps (Figure 1). The installation site is a flat area, at an elevation of $\approx 2000$ m a.s.l. surrounded by peaks rising up to $\approx 3000$ m. On Feb. 5, 2016, a snow slab avalanche released from Chlein Sattelhorn, and was recorded both by the seismic and by the infrasound array (Figure 1, Figure 2a).

The seismic array (Seismic Instruments Inc.) consisted of 7 elements deployed with a circular geometry and maximum aperture (maximum distance between 2 array elements) of 75 m (Figure 2c). The array was equipped with vertical geophones with a corner frequency of 4.5 Hz and a sensitivity of 28.8 V/m/s. The geophones were attached with anchors to large rocks on the ground and subsequently buried by snow, which substantially reduced the effect of wind noise. Seismic data were sampled at each geophone at 500 Hz and 24 bits precision. Data were recorded locally at the central acquisition system. The entire system was powered with solar panels and batteries, and the total power requirement was approximately 7 W.

The infrasound array (FIBRA; www.item-geophysics.it) operated with fiber optic connection among the 5 different array elements. The fiber optic connection allows to increase significantly the signal to noise ratio, and prevents the risk of damage related to lightning or electric discharges. The array was deployed following a triangular geometry, with two central elements, and had a maximum aperture of 160 m (Figure 2d). Each array element was equipped with a differential pressure transducer (prs025a), with a sensitivity of 400 mV/Pa in the pressure range of +/- 12.5 Pa, and a frequency response between 0.01 and 200 Hz. Analogue pressure data were converted to digital at each array element, with a sampling rate of 50 Hz and 16 bits dynamics, and were transmitted trough fiber optic to a central unit, where data were synchronized with GPS timing. Data were both recorded locally and made available trough TCP/IP for data transmission. Power requirement was $\approx 1$ W for the central unit and as low as $\approx 0.1$ W for the array element.

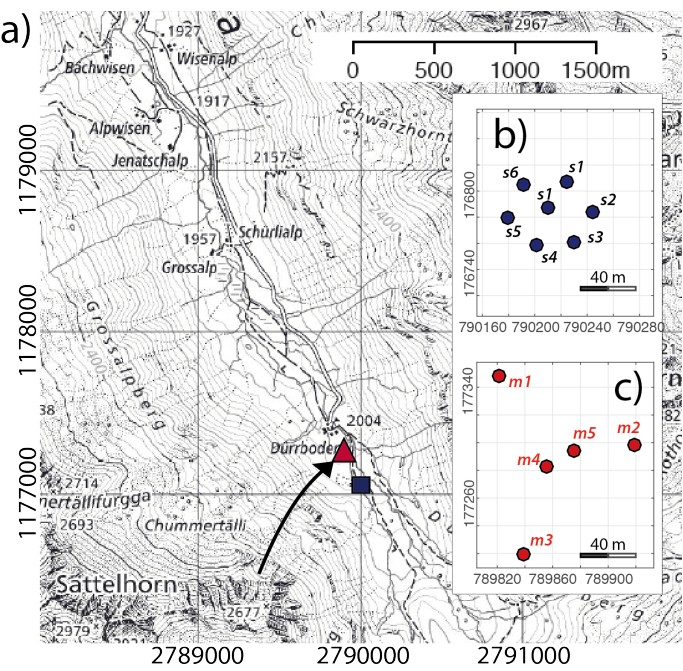

**Figure 1.** Map (a) showing the location of the Dischma valley, south of Davos, Switzerland. The location of the infrasound array (red triangle) and the seismic array (white square) are show, as well as the Chlein Sattelhorn avalanche path (black arrow). Positions are given in Swiss coordinates (CH1903). Reproduced by permission of swisstopo (JA100118). Details of the geometry of the seismic (b) and infrasound array (c).

The data recorded by the two arrays were synchronized by comparing the timing of local and regional earthquakes recorded by the infrasound as well as the seismic array. This guarantees a timing accuracy of $< 2$ seconds, which is sufficient for the seismo-acoustic comparison presented here.

The study site was also equipped with automatic cameras collecting images every ten minutes, used to visually monitor the activity on the slopes surrounding the arrays. The camera system was colocated with the central element of the infrasound array.

## 3 The dry-snow avalanche of 5 February 2016

In the morning of 5 February 2016, at 05:18 UT, a medium sized dry-snow avalanche released from Chlein Sattelhorn (Figure 2b), at an elevation of $\approx 2600$ m. The avalanche traveled a distance of 1200 meters and stopped at the bottom of the Dischma valley, at an elevation of $\approx 2030$ m at a short distance ($<100$ m) from the infrasound array (Figure 3). The event occurred during a snow storm. Nevertheless, based on the images from the automatic cameras we confirmed that the avalanche released between 4 February 2016 at 17:40 UT and 6 February 2016 at 07:40 UT. The avalanche deposit was first clearly visible on the morning of Feb. 6 (08:30 UT), when the weather weather cleared (Figure 3).





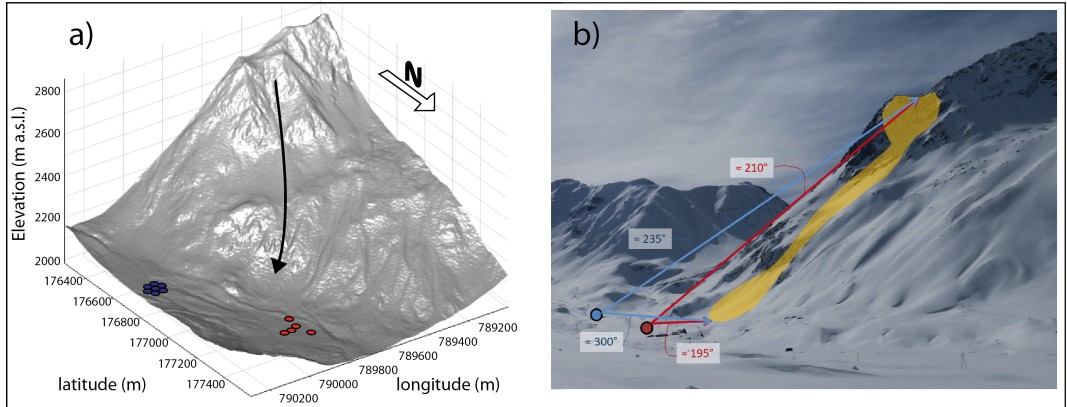

**Figure 2.** (a) Digital Elevation Model showing the installation site within the Dischma valley, south of Davos, Switzerland, with the position of the seismic array (blue dots), the infrasound array (red dots) and the Chlein Sattelhorn avalanche path (black arrow). (b) Photo of the field site with the position of the seismic array (blue dot), the infrasound array (red dot) and the approximate contour of the Chlein Sattelhorn avalanche from 5 February 2016 (orange). The approximate backazimuth angles to the start zone and runout zone of the avalanche relative to the seismic and infrasound array are also shown (colored arrows).

The flow characteristics and evolution (flow depth and velocity) were reconstructed using the RAMMS model (Figure 4)
(*Christen et al.*, 2010). We used RAMMS::Avalanche (version 1.7.20) for the simulations of Chlein Sattelhorn. The model requires a detailed digital elevation model as well as an estimate of the initial release volume, i.e. an initial release area and a fracture depth. The inital digital elevation model (DEM) is the swissAlti3D DEM (2 m grid resolution). For the simulation, we did a bilinear interpolation to 5 m. The release volume (with release depth of 80 cm) was 9.525 $m^3$. We used calibrated friction values for small avalanches, with a return period of 10 years. The modeled flow depth evolution (*Marchetti et al.*,
2019), predicts a total flow duration of $\approx$ 90 seconds, with $\approx$ 60 seconds required by the avalanche to initiate, accelerate, and reach the valley bottom, followed by $\approx$ 30 seconds of snow deposition. Since the path geometry is characterized by a sharp terrain break at an elevation of approximately 2300 m (Figure 4c), the modeled avalanche accelerated along the release area with slopes exceeding 35 degrees, rapidly decelerated and lost mass at the terrain break (Figure 4d). The modeled avalanche then accelerated again after entering a steep (slope > 30 degrees), narrow channel (< 50 m), within the lowest part of the
path. Finally, the flow slowed down when it reached the valley bottom at an elevation of $\approx$ 2030 m (Figures 2,4), where the snow mass was spread out horizontally on the runout area. The modeled snow avalanche qualitatively compared well to the information we obtained from the images from the automatic cameras (Figure 2b).

RAMMS predicted a maximum flow depth of almost 3.5 m, that was reached after a travel distance of $\approx$ 200 m along the path. The maximum front velocity, of $\approx$ 35 m/s, was reached at the end of the first, and highest, part of the path, before the
deceleration at the terrain break (Figure 4). Lower values of front velocity and flow depth result from the model below the terrain break.



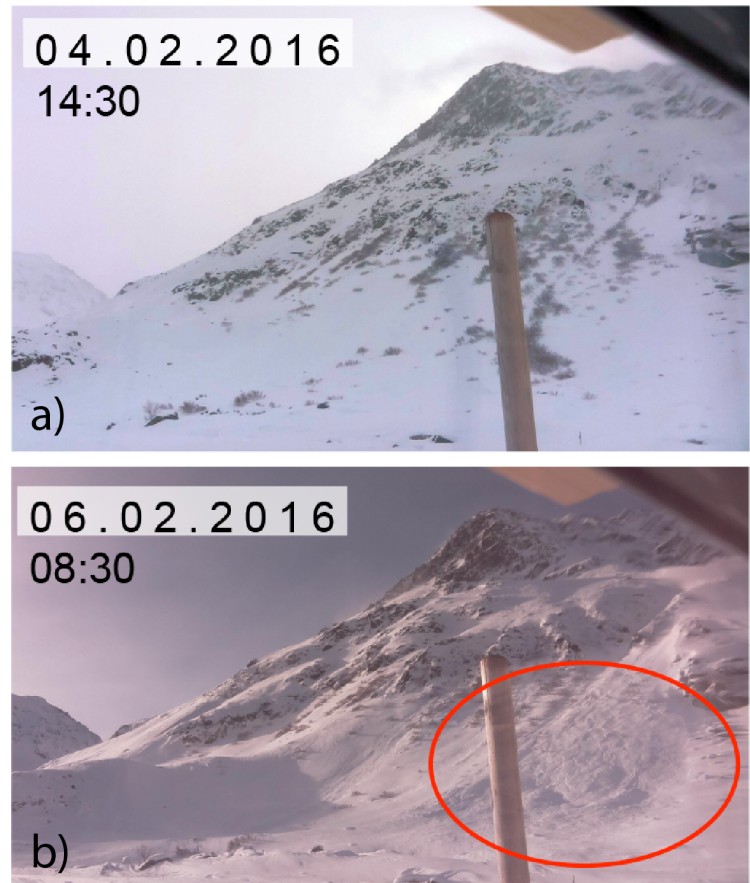

**Figure 3.** Picture showing the slope west from the array in the afternoon of February, 2nd (a, last clear image before the event) and in the morning of February, 6th, 2016 (b, first clear image after the event) that shows the avalanche accumulation area.

## 4   Seismo-acoustic records of the event

The event from 5 February 2016 was clearly recorded by the seismic and infrasound arrays (Figure 5a, b) (*Marchetti et al.*, 2019). Both signals consisted of two distinct phases, according with the flow evolution modeled by RAMMS (Figure 4). These
two phases appear to be controlled by the path geometry forcing the avalanche to slow down and loose mass at the terrain break at an elevation of 2300 m.

The seismic signal has an emergent waveform and a duration of $\approx 60$ s. It is characterized by two phases of similar amplitude ($1.5 \ 10^{-6}$ m/s), peaking around 05:18:50 and 05:19:20 UT. The signal spectrum shows energy mostly confined between 3.5 and 12 Hz, with the peak frequency around 6 Hz. The frequency response of the geophones limits the spectral analysis to
frequencies $> 4.5$ Hz (Figure 5c), therefore we cannot exclude lower frequency components.



**Figure 4.** Figure showing the flow extent modeled by RAMMS and highlighting maximum flow depth (a) and flow velocity (b). Elevation of the avalanche path (c) as a function of the horizontal distance with respect to the central element of the infrasound array. (d) Profiles of the modeled flow depth (blue) and flow velocity (red) along the path.

The infrasound record of the event is shorter ($\approx 35$ sec), has a similar emergent waveform, with two sparate phases reaching a maximum amplitude of $\approx 0.5$ Pa (Figure 5b). The spectral energy of the infrasound signal is wider, spanning between 0.5 and 8 Hz, with a clear peak at $\approx 3.3$ Hz (Figure 5c).

The infrasound and seismic data were processed by applying a multichannel correlation analysis, to identify signal from

noise in terms of signal back-azimuth and apparent velocity. The procedure, described in detail by *Ulivieri et al.* (2011), identifies coherent data recorded within a given time window assuming planar wavefront propagation. Once a coherent signal





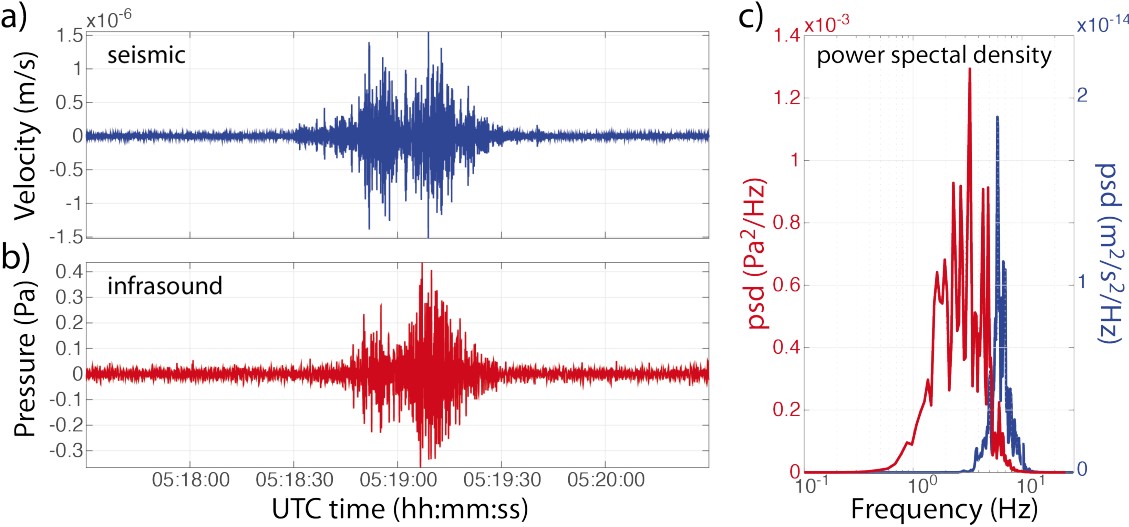

**Figure 5.** Seismic (a) and infrasound (b) record of the avalanche that occurred on 05 February 2016. c) Power Spectral Density (psd) of the seismic (blue) and infrasound signal (red).

is identified, based on signal correlation threhsold ($> 70$ %), the corresponding back-azimuth ($Baz$) is calculated. The back-azimuth corresponds to the propagation angle from the array to the source, measured with respect to the geographic North in the horizontal plane of the array. Once the back-azimuth is identified, the apparent velocity ($c_a$) is calculated, as the ratio

between the real propagation velocity ($c$) and the sin of the take off angle ($c_a = c / \sin \theta$). The apparent velocity corresponds to the velocity the wave would have if it was traveling in the plane of the array, and increases for higher elevation sources. It would be infinite for a source located directly above the array, as all the elements of the array would record the signal simultaneously.

In the specific case of snow avalanches, i.e. a source moving downhill, the apparent velocity is expected to decrease with time (*Marchetti et al.*, 2015). Similarly, the back-azimuth is expected to change with time. This aspect has been used to identify

snow avalanches from other sources with infrasound array analysis, allowing for real-time monitoring and identification of snow avalanches at source-to-receiver distances of several km (*Marchetti et al.*, 2015; *Ulivieri et al.*, 2011; *Mayer et al.*, 2018).

For the avalanche from 5 Ferbruary 2016, the multichannel correlation analysis was performed over time windows of 5 seconds, and with an overlap of 4.5 seconds, both for the seismic and for the infrasound data. According to the recorded frequency spectrum (Figure 5c), infrasound and seismic data were bandpass filtered between 1 and 10 Hz. The event then

appears as a cluster of detections (Figure 6), each associated with a corresponding value of back-azimuth ($Baz$) and apparent velocity ($c_a$), calculated for each signal time window. From Figure 6, it is clear how infrasound signal starts to be detected at 05:18:50 UTC, with a back-azimuth of 211 degrees N and an apparent velocity of 365 m/s. The apparent velocity decreases constantly down to 327 m/s, until 05:19:15, while the back-azimuth remains quite stable for the first 15 seconds of the recorded signal, until 05:19:05, and decreases down to 200 degrees N afterwards. Such a variation of apparent velocity and back-

azimuth is consistent with the path followed by the observed event, that moves initially North/North-West towards the array





(20 degrees N, back-azimuth from the array 210 degrees N), to turn clockwise along the path when the flow moves downhill and approaches the array (Figure 3). Between 05:19:15 and 05:19:25 UTC, the back-azimuth is quite stable, while the apparent velocity increases up to $\approx$ 350 m/s. We interpret this variation as an artifact, resulting from the short distance between the accumulation area and the array ($<$ 200 m), thereby violating the planar wavefront assumption.

Unlike infrasound, that has a duration of 35 seconds and is marked by a clear variation of wave parameters (back-azimuth and apparent velocity), the seismic signal radiated by the event is much longer in duration ($\approx$ 60 sec), and changes in wave parameters were less clear. The first seismic detections were recorded around 05:18:40, $\approx$ 10 seconds before the first infrasound detection, with a back-azimuth values between 220 and 250 degrees, corresponding reasonably well with the release area of the snow avalanches (Figure 1). During the following 20 seconds we observe a general migration of the seismic back-azimuth,

up to $\approx$ 270-300 degrees N at 05:19:00 UT, corresponding to the runout area. Afterwards, the seismic back-azimuth remains rather stable until the end of the event at 05:19:45 UT.

Considering the apparent velocity, the array processing highlights high values ($>$ 650 m/s) at the beginning and at the end of the event. These values are in agreement with phase velocities (500-950 m/s) measured by *Vilajosana et al.* (2007a) for snow avalanches in Ryggfonn in Norway, as well as values used by *Lacroix et al.* (2012) for beamforming in the French Alps.

The central part of the signal, between 05:19:00 and 05:19:15 UT, is characterized by a lower propagation velocity ($\approx$ 330 m/s), suggesting that the seismic array is likely recording infrasound waves. This corresponds to the time when the infrasound amplitude was maximum (Figure 6a). We suggest that the central part of the signal is strongly affected by the infrasound radiated by the event, that converts to seismic waves at the earth free surface and is efficiently recorded by seismometers. This is an agreement with results obtained by *Heck et al.* (2017) for an avalanche that did occur from the same path in 2017, and

applying the multiple signal classification (MUSIC) analysis to seismic array data.

## 5   Elastic energy radiation along the avalanche path

The results of infrasound and seismic array processing presented in Figure 6, allow us to describe the mutual infrasound and seismic energy radiation during the avalanche. Just considering the event duration, it is clear from Figure 6, that the avalanche initiation phase is radiating seismic energy in the ground, while no or minor infrasound is radiated into the atmosphere. This is

likely related to the first stage of the event, whilst the powder front is not developed yet. Only 20-25 seconds after the avalanche onset, once the flow accelerates (see auxiliary material), infrasound is radiated from a source that is moving downhill along the avalanche path, as tracked by the infrasound wave parameters (back-azimuth and apparent velocity, Figure 6c, e).

Following the approach described by *Marchetti et al.* (2015), we use the back-azimuth and the apparent velocity of the seismic and the infrasound detections to investigate the position along the avalanche path where the different portions of the

signals are generated. For each point of the DEM, we calculate the corresponding values of back-azimuth from the seismic ($Baz_s$) and infrasound ($Baz_i$) arrays, and the expected values of apparent velocity ($c_a$) of the recorded infrasound wavefield (Figure 7). We obtain a map of theoretical values, showing the expected values for the seismic and infrasound wave parameters

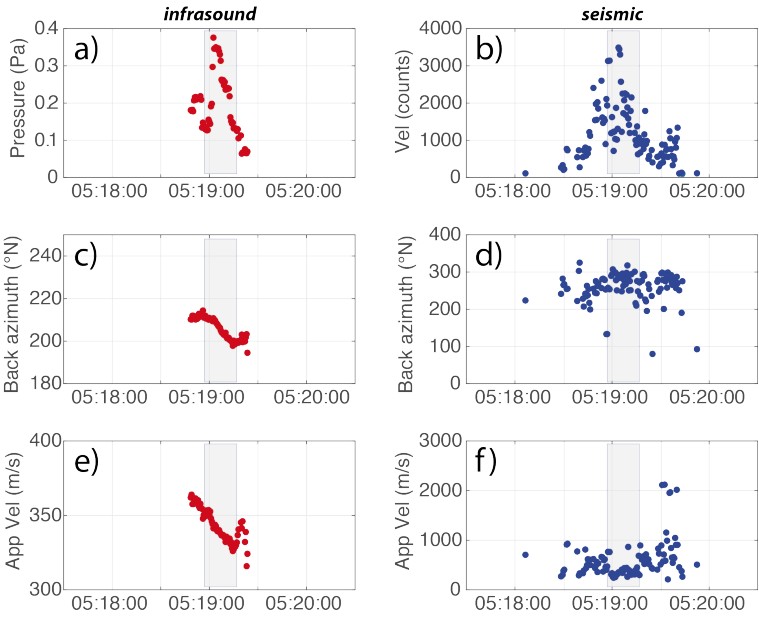

**Figure 6.** Amplitude, back-azimuth and apparent velocity of infrasound (a,c and e, respectively) and seismic (b,d and f, respectively) detections for the avalanche of 5 February 2016. The shaded area highlights the time window of sound propagation velocity recorded for the seismic signal.

produced by any point source located on the DEM. A point source moving along any path on the topography would therefore be recorded with values of back-azimuth and apparent velocity, varying according to the theoretical values.

The back-azimuth provides an estimate of the source position in the horizontal plane defined by the array, while apparent velocity is reflecting the source elevation. As expected, the map shows that the back-azimuth varies between 0 and 360 degrees around the seismic and infrasound arrays (Figure 7a, b), independently of the elevation and the distance from the array. The infrasound apparent velocity changes, according to the local topography, from a minimum of 330 m/s up to a maximum value of 400 m/s (Figure 7c), solely affected by the distance from the array and the absolute elevation. Therefore, a proxy of the

3d source position requires combining back-azimuth and apparent velocity. We account simultaneously for back-azimuth and apparent velocity by calculating for each point of the DEM the product ($BV = Baz_i$ x $c_a$) of theoretical values (Figure 7d). The resulting parameter defines a new map, with values depending from both the planar position and source elevation. Such an approach can be easily applied to infrasound wave parameters, while the use of the apparent velocity derived for the seismic wavefield appears complicated by variable phases and complex source-to-receiver travel paths.

Once the theoretical values of the seismic ($Baz_s$) and the infrasound ($Baz_i$) back-azimuth, the infrasound apparent velocity ($c_a$) and their product ($BV$) are evaluated (Figure 7), the source radiating areas for the seismic and infrasound signals can be evaluated from real detections (Figure 6.) We performed a blind search to minimize the difference between wave parameters ($Baz_s$, $Baz_i$, $c_a$ and $BV$) calculated for the seismic and infrasound detections and the theoretical values calculated for the



Earth **Surface**
**Dynamics**
Discussions



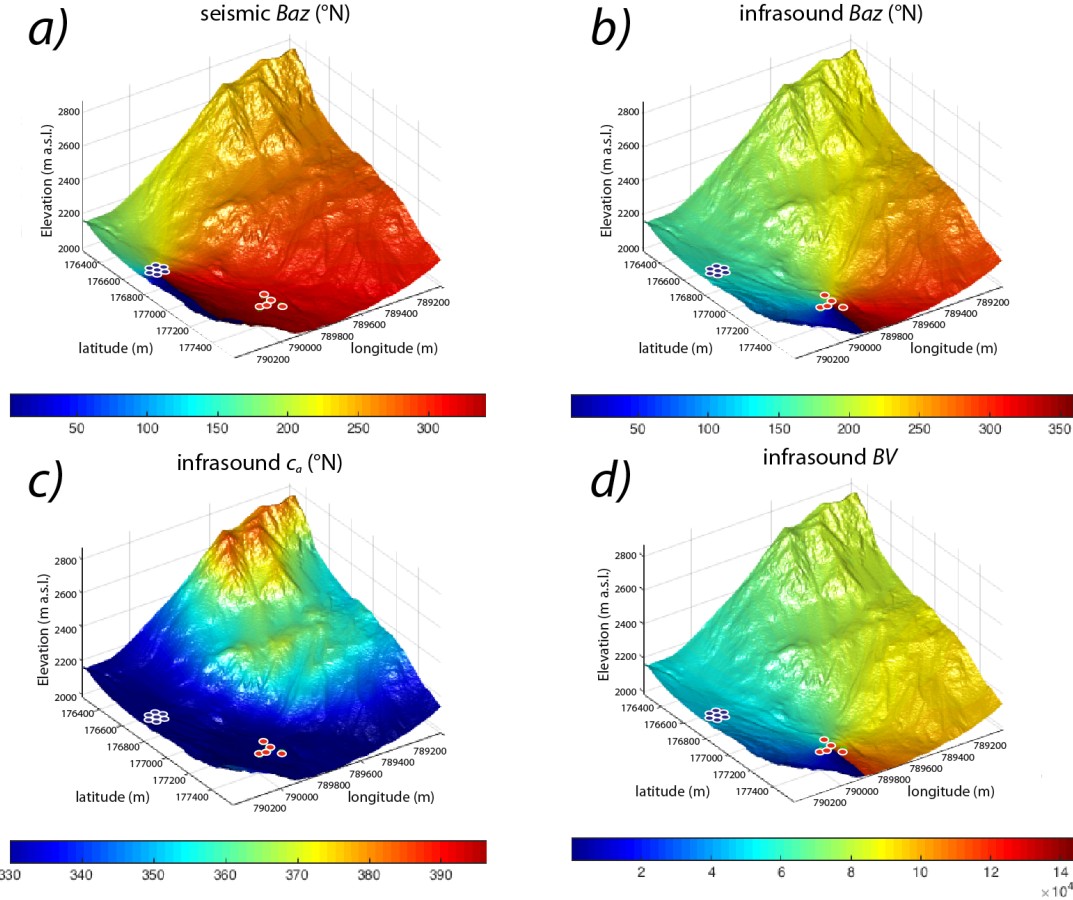

**Figure 7.** Theoretical values of back-azimuth at the seismic array ($Baz_s$, a), back-azimuth ($Baz_i$, b) and apparent velocity ($c_a$, c) at the infrasound array for any point source of seismic and infrasound energy located on the DEM. Product between back-azimuth and apparent velocity of infrasound wavefield (d). The position of the infrasound and seismic arrays are shown by red and blue circles respectively.

DEM. Figure (8) shows all the possible source points along the DEM based on the seismic back-azimuth (Figure 8a), the
infrasound back-azimuth (Figure 8b), the infrasound apparent velocity (Figure 8c) and their product (Figure 8d). The dark-red areas highlighting all the points of the DEM satisfying a minimum difference threshold. Figure 8 shows that, considering only one parameter at once, only a limited information on the source radiation area can be deduced, unless a constraint of the avalanche path is applied. Such an approach, was applied successfully in previous studies that evaluated the avalanche velocity from infrasound detections (*Havens et al.*, 2014; *Marchetti et al.*, 2015), but limits the analysis to a single avalanche path.
Considering the seismic back-azimuth ($Baz_s$, Figure 8a) only, for example, the detections do not provide any constraint on the source position, as they are consistent with many different directions around the array spanning between 200 and 325 degrees N. However, if we assume that the seismic source is confined within the avalanche path (Figure 2a), it appears that





the seismic energy is radiated from the detachment point to the depositional area. Moreover, the scattered values of the back-azimuth of the recorded seismic signals, suggest that multiple sources of seismic energy are active at the same time in different

sectors of the avalanche path.

The relative position of the avalanche path and the infrasound array, almost in line, influences the efficiency of the infrasound back-azimuth to identify the source position along the path. Considering the infrasound back-azimuth alone ($Baz_i$, Figure 8b), the back-projection of infrasound detections to the topography does no allow us to constrain the position of the source along the path. The infrasound apparent velocity constrains the source elevation ($c_a$, Figure 8c). The maximum value of the apparent

velocity of 364 m/s (Figure 6) limits the energy radiation to the lowest part of the avalanche path, clearly suggesting that no, or minor, infrasound is produced high up in the path during the initiation phase. This conclusion is confirmed by the blind search of the infrasound energy radiation area, based on the combination of back-azimuth and apparent velocity ($BV$, Figure 8d). Here, the minimization of residuals between theoretical and measured values, highlights a limited area on the entire DEM, from the base of the starting zone, where the avalanche accelerates and follows the channel down to the valley.

The clear variation of the back-azimuth and apparent velocity of the recorded infrasound signals also provides a strong constraint on the source mechanism of infrasound energy. As discussed in detail by *Marchetti et al.* (2015), in case of multiple sources, the array analysis identifies the most energetic source. Therefore, our results suggest that snow avalanches are characterized by a dominant source of infrasound energy, likely the powder front, allowing to treat the source mechanism of infrasound mostly as a point source moving downhill.

The location of infrasound and seismic energy radiation along the avalanche path presented in this study, is in agreement with the hypothesis that infrasound is produced by the power cloud and with the dynamical evolution of a PSA in terms of an eruption current (*Carrol et al.*, 2013). They showed that the powder cloud formation is strongly enhanced by the narrowing of the avalanche path, while it is limited by the path spreading in the initiation and deposition area. Our seismic and infrasound array observation, clearly shows that while seismic energy is radiated as an elongated source all along the avalanche path, the

infrasound signal is radiated mostly from the powder cloud, that develops only within the narrow avalanche channel and is missing in the wider starting and deposition areas (*Carrol et al.*, 2013).

## 6   Discussion

Our results show that seismic and infrasound energies are radiated in different parts of the avalanche path. Seismic energy is radiated all along the path, while infrasound is radiated when the flow is accelerated within the channel. Moreover, the

observed trend of back-azimuth suggests that infrasound is likely produced by the downhill moving powder cloud. The cloud can be approximated as a point source, which radiates infrasound from a single position at a given time. Differently, the scattered back-azimuth of seismic detections suggests that seismic signals are most likely produced by multiple sources, or by an elongated source, acting along the path at the same time (Figure 6).

Previous studies (*Surinach et al.*, 2001; *Vilajosana et al.*, 2007b) showed that seismic signals produced by snow avalanches

mostly consist of surface waves. They assumed that the seismic energy was radiated mostly by the basal friction and snow



Earth **Surface**
**Dynamics**
Discussions



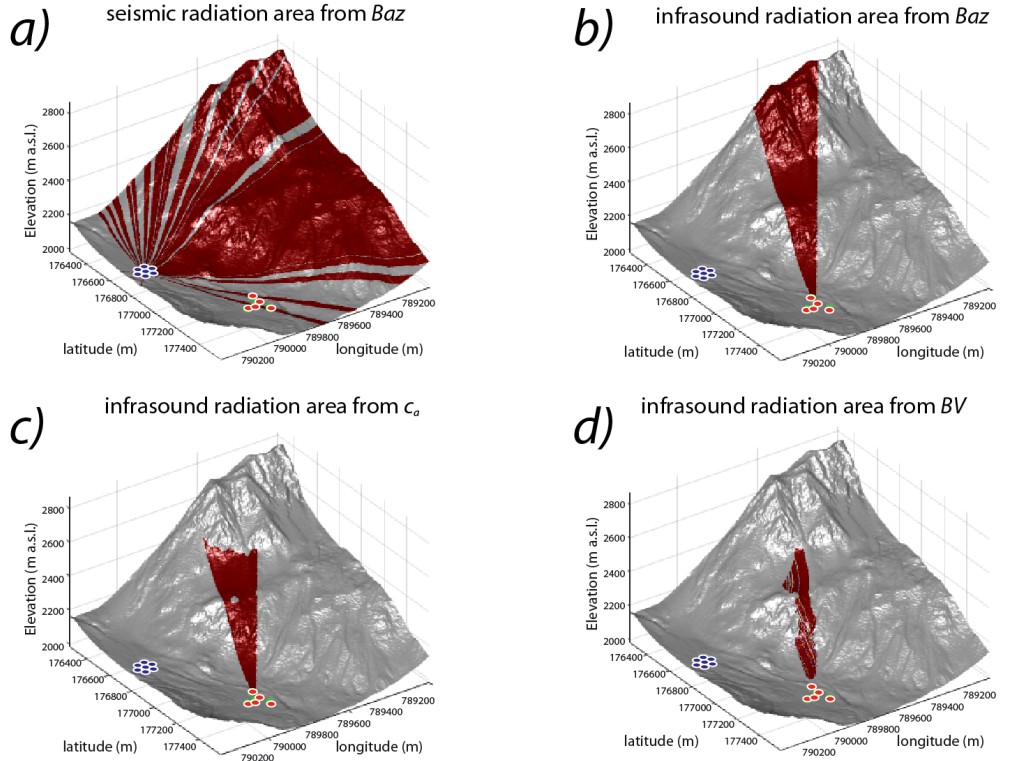

**Figure 8.** Possible radiation areas (dark red) of seismic and infrasound energy, obtained from seismic back-azimuth ($Baz_s$, a), infrasound back-azimuth ($Baz_i$, b), apparent velocity ($c_a$, c) and the combination of infrasound back-azimuth and apparent velocity ($BV$, d).

ploughing at the avalanche front (*Vilajosana et al.*, 2007b). The seismic energy was calculated accounting for geometrical spreading and attenuation of surface waves along the front-to receiver distance. Any possible contribution from multiple sources along the path or by an elongated source were not considered. This could lead to an underestimation of the total seismic energy of the event.

An additional constraint to be considered is that the recorded seismic amplitude is possibly increased by the sound transmitted locally into the ground. In this study, this process is confirmed by the combined analysis of seismic and infrasound detections (Figure 6). During the phase of maximum infrasound radiation, the recorded seismic signal is propagating at the velocity of sound (shaded area in Figure 6). This indicates that infrasound energy is transmitted locally to the ground and recorded with seismometers. Sound propagation velocities of seismic signals produced by snow avalanches were reported also

for previous seismic array investigations (*Heck et al.*, 2017; *Lacroix et al.*, 2012).





The process of infrasound to seismic energy transition was described by *Ichihara et al.* (2012). An infrasonic wave hitting the ground ($p(t,x)$) produces a vertical ground velocity ($w(t,x)$) that is directly proportional to the amplitude of the incident wave ($w(t,x) = Hp(t,x)$). The conversion factor ($H$) is defined as:

$$H = \frac{\exp(\frac{-i\pi}{2})c}{2(\lambda+\mu)} \frac{\lambda+2\mu}{\mu},$$
(1)

where $\lambda$ and $\mu$ are Lame's constants of the ground and $c$ is the velocity of sound. Considering typical values of the Lame's
constant for soil ($\approx 10^8$) Pa, and a sound velocity ($c$) of 340 m/s, the conversion factor ($H$) results $\approx 5x10^{-7}$ m/s/Pa. Therefore, an infrasonic wave of 1 Pa will produce a detectable seismic signal in the ground. In the specific case of the 5 February avalanche (Figure 5), the recorded pressure of 0.4 Pa, will produce a seismic signal with an amplitude of $\approx 3x10^{-7}$ m/s, that corresponds to $\approx 20$ % of the recorded seismic amplitude. This is in agreement with sound velocities recorded in the seismic data during the phase of maximum infrasound radiation (Figure 6).

This study, combining for the first time seismic and infrasound array data, highlights the complexity of the seismic radiation by snow avalanches and the contribution of the air-to-ground energy transmission. These have an influence on the recorded seismic signal and, if not accounted for, might limit the applicability of seismic signals for energy estimations. The absolute seismic amplitude, and corresponding energy, can change according to snow characteristics (dry/wet) (*Vilajosana et al.*, 2007b), and efficiency of air-to-ground energy transmission (*Ichihara et al.*, 2012). This approach is even more critical considering that
seismic energy is radiated all along the avalanche path (Figure 8a). Moreover, it requires a-priori characterization of the quality factor of surface waves at the site (*Vilajosana et al.*, 2007b), thus preventing a general application of the proposed procedure at various sites.

Similarly, infrasound amplitude is expected to change dramatically as a function of avalanche type (dry/wet) and path geometry, and our results suggest that estimating avalanche size from infrasound signals could be difficult. Signal duration
is, for example, reflecting only the part of the path where the avalanche is accelerated, or where the powder cloud develops (Figure 8d). Considering the radiation of sound by a moving body assumed to be a solid sphere, *Naugolnykh and Bedard* (1990) suggested that the frequency of recorded infrasound must scale inversely with the body size as follow:

$$f = c/\pi D,$$
(2)

where $c$ is the velocity of sound in the atmosphere while $D$ is the diameter of the sphere.

For the specific case of the avalanche recorded on 5 February 2016, eq 2 predicts a moving sphere-like body with diameter
$D$ of $\approx 30$ m. This value is obtained by assuming a sound propagation velocity of 330 m/s and considering a peak frequency of 3.3 Hz (Figure 5), and is of the same order as the width of the avalanche channel (<50 m). Nevertheless, a snow avalanche is far from being a rigid sphere moving in the atmosphere. Already *Naugolnykh and Bedard* (1990), suggested that additional processes might contribute to the avalanche infrasound radiation, such as the turbulent pressure pulsation of the powder cloud and/or secondary source mechanisms. Therefore, while the approach proposed by *Naugolnykh and Bedard* (1990) seems





working fine as a first approximation, analyses will be required to further investigate the source mechanisms of infrasound possibly combining infrasound, seismic and high resolution video observation.

## 7 Conclusions

Results presented here, and obtained from seismic and infrasound array analysis, highlight two separate mechanisms of elastic energy radiation by a snow avalanche. The infrasound energy is radiated only when the powder part develops, and is not
produced during the initiation or deposition phase. The duration of the infrasound signal is thus not representative of the entire volume of snow that was transported by the avalanche. Because of the clear migration of infrasound detections in terms of back-azimuth and apparent velocity, we suggest that the source mechanism can be interpreted as a moving point source. The clear wave parameters obtained from the array analysis, suggest that infrasound can be used as an efficient monitoring for avalanche detection purposes. Back-projection of the infrasound detections on the avalanche path, suggested that the infrasound energy is
radiated only when the flow is confined within a narrow path. According to the analytical formulation of *Carrol et al.* (2013), such a condition enhances the formation of the powder front.

The seismic signal is, instead, produced during the entire avalanche evolution, including the initiation and deposition area. Therefore, the signal duration is longer and more representative of the entire flow evolution and run-out distance. Unlike for infrasound, the seismic back-azimuth and apparent velocity values were more scattered, and this makes the detection and
location of avalanche events less straightforward than with infrasound. Furthermore, the scattering of wave parameters suggests multiple sources that act simultaneously along the path.

In agreement with *Heck et al.* (2017), the combined seismic and infrasound array analysis, showed also that during the phase of maximum infrasound radiation, seismic energy is strongly affected by the infrasonic signal. This needs to be accounted for, when the seismic amplitude is used to estimate the avalanche energy. Similarly, the amplitude of recorded infrasound is
controlled by the avalanche type (wet/dry) and the flow evolution (i.e. the formation of the powder cloud). Good results are obtained, for the avalanche event investigated here, considering the frequency of the recorded infrasonic signal, and assuming the source as being produced by a moving sphere (*Naugolnykh and Bedard*, 1990). For the specific case of the avalanche of 5 February 2016, the recorded peak infrasound frequency of 3.3 Hz is consistent with a sphere like body with a diameter of 30 m, in agreement with the the geometry and extension of the avalanche path.
Although many open questions remain concerning the mechanisms of infrasound and seismic energy radiation by snow avalanches, the combined seismic and infrasound array analyses presented in this study helped in clarifying some key aspects of the recorded seismic and infrasound signals, like source origin, possible source mechanisms and mutual relation. Further studies will be required, however, to investigate in detail the source mechanisms of elastic anergy radiation, secondary source processes, like turbulence of the powder front, and possible use of the seismic and the infrasound signal to evaluate the magni-
tude of the event.





*Data availability.* Infrasound and seismic detections resulting from array processing and used here to achieve most of the findings and create most of the figures are freely available in the Open Science Framework repository (https://osf.io/p28gc/), doi:10.17605/OSF.IO/P28GC.

*Video supplement.* RAMMS::Avalanche simulation, depicting the flow depth (m) along the avalanche path. Red colors indicate flow depths larger than 2m.

*Author contributions.* All of the authors contributed actively to the manuscript. In detail, E.M. and A.v.H. worked on the infrasound and seismic data array analysis, developed the model of infrasound and seismic energy partitioning and wrote to the text. M.C. performed the RAMMS modeling of the avalanche. M.C.S. and G.B. contributed to the text, figure and discussion. M.C.S. worked primarily on seismic data while G.B. worked mostly on infrasound data and developed the relationship between infrasound and volume.

*Competing interests.* No competing interests are present.

*Acknowledgements.* Part of this research was supported by a grant of the Swiss National Science Foundation (200021149329).



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
