# Peer review of "Seismo-acoustic energy partitioning of a powder snow avalanche"

_Earth Surface Dynamics, 2019_

## Referee Comment (RC1) · Emma Surinach (Referee) · 31 Dec 2019

The study combines the seismic and infrasound data obtained in two arrays distant < 500 m placed at the end of the powder-snow avalanche path occurred on February 5, 2016. The infrasound array was in a direct visual position of the avalanche path, while the seismic array not. The study is limited to the frequency of 4.5-10 Hz for the geophone data and to 1-10 Hz for the pressure data measurements. The results are contrasted with the outputs of RAMMS modeling. Conclusions refer, basically, to the different origin of the infrasound and seismic wavefields and also to the implication in the calculation of the avalanche size. It is a very interesting contribution, although the authors have not take into account the previous paper of Kogelnig et al., (2011) where seismic and infrasound time series are compared for four different types of avalanches

at the VdLS experimental site, descending along different paths, unlike the one presented in the manuscript under review in which only one avalanche is studied. In this paper the infrasound and seismic time series obtained in collocated sensors with a common time are compared, and also with the time series obtained in two more seismometers placed along the avalanche path. Additionally, a comparison was made with other "in situ" direct measurements (flow depth and internal velocities). The frequencies involved in the study are in the range of [1-40] Hz for both type of measurements. Of interest is the content of "low" [1-3] Hz frequencies of the seismic when comparing with the infrasound. Because of the completeness of these data, with respect to that of the data of the manuscript under review, the authors must take into account in their discussion and conclusions the results obtained previously. In principle, part of the obtained results by the authors could confirm the previous ones or contradict them. The use of the combination of the two arrays in this study is very positive, but the authors must be aware of the limitations of their study. In addition, the results presented also depend on the specific topography. One of the difficulties of the comparison of the results of the two arrays is that the infrasound array has a direct view of the avalanche flow and the array of geophones does not. What would happen in the case of the existence of a shadow zone for the infrasound? Or if the seismic array had been collocated with the infrasound array?

As regards Section 5. Kogelnig et al. (2011) includes a section dedicated to the source of infrasound and seismic signals. There, a synthetic signal is obtained using the expression of Ffowcs Williams (1963) that describes the acoustic intensity generated by a turbulent source in motion. The modeling results are compared with the infrasound time series obtained from an avalanche. In addition, due to the existence of a suspension layer that can generate infrasound, an explanation is included for not considering a unic specific source of infrasound. In addition, the flow dimension D is calculated for the dominant frequencies. The authors must take into account in their discussion and conclusions the results obtained previously.

A remark on Figure 4. This figure is very important in the interpretation of the time series and the results. Note that the origin of the time series corresponds to the farthest distances. To facilitate interpretation with the time series, the authors must convert the distances into time (using the obtained speeds) and reverse the origin of the distance. In addition, the slope angle (derived from the profile) incorporated in Figure 4d with the outputs of the RAMMS model will help to better correlate the slope change with the features of the time series.

The manuscript is of interest to the community. It contributes in one more step to the knowledge of the wave field generated by snow avalanches for the application to its detection. I recommend to the authors an exhaustive review taking into account my comments. Below further details. In my review I remark when Kogelnig et al. (2011) also had a contribution, consider it.

The limitation in the frequency content used in the study must be indicated in the abstract.

- Line 48. Please, confirm that this reference is correct. There are different contributions of these authors with the same title, e.g.

.Naugolnykh K, Bedard A (2001) A model of the avalanche infrasound radiation. In: Proceedings of the 24th Canadian Symposium of Remote Sensing. pp 871–872

.Naugolnykh K, Bedard A (2002) IEEE International Geoscience and Remote Sensing Symposium DOI: 10.1109/IGARSS.2002.1025713.

But I am unable to find that you mentioned:

.Naugolnykh, K., and Bedard, A.: Model of the avalanche infrasound radiation. Proceeding of International Snow Science Workshop, Jackson, WY, 19-24 September 2004, 871-872, 1990.

Lines 76- 90. -Line 77. Please, check the figure numbers. e.g. Is figure 2c correct or it is 1b?

Although it is indicated in the abstract you must mention here the distance between the arrays -Line 85. Figure 1c?

-Lines 91-93. This assertion will be correct assuming that the earthquake is recorded in the infrasound sensors. An explanation on this, references, or more detail is needed. ... <2 s assuming the difference in wave travel time and wave propagation speeds of ... and a distance of ... -Line 100. Please, check if Figure 3 is correct.

- Line 123. Indicate which sensor corresponds to the time series presented in Figures 5 a and b. Or are they stacked time series?

- Line 131. Are you sure about 35 s? Could you specify the signal limits here even if you do it below?

- Lines 140-143. In fact, there are two speeds, one of the infrasound waves in the air and the other corresponding to the source (avalanche). Authors should specify this somewhere, here or above in the presentation of the method.

- Line 147 February

- Line 149. Note that the only effect of the low pass filter is in the infrasound, because the geophones natural frequency is 4.5 Hz

- Line 160 approx. 35 s as indicated in Line 131.

- Line 162. An explanation of the difference between the detection of time arrival of the matrix and that of the seismic amplitudes observed at 18.30 (Figure 5a) that are clearly due to the avalanche is necessary. Given that the velocity of the seismic waves in relation to the avalanche speed and that of the infrasound in the air, it seems that the avalanche started earlier than indicated.

- Lines 167-169. Does it refer to the processing of the seismic array or that of infrasound? Please specify. Vilajosana et al. (2007a) obtained the mentioned ground phase velocities from waves generated by explosions at Ryggfonn. These speeds are

[Figure]

**ESurfD**

independent of avalanches. This is a feature of the site. I think there is a misunderstanding. Please clarify.

- Line 176. Section 5. See previous comment on this.

- Line 181. Auxiliary material. Do you mean the video? Include the reference.

- Line 181. ...radiated from a point source

- Line 184. ... along the path considering a point source

- Lines 210-211. This could be an effect of the relative position of the arrays as mentioned by the authors in Line 214.

- Line 227. With your results, there is not enough information to generalize to all the avalanches, in plural. - Line 232. Remember the content in Kogelnig et al., (2011).

- Line 246. and references therein....

- Line 249. Note that the effect of filtering from 1 to 10 Hz and realistically, from 4.5 to 10 Hz, also presents problems in the quantification of the energy, since part of the signal is lost.

- Line 252. This was also mentioned in Kogelnig et al., (2011)

- Line 260-264. The authors must specify that this is in the range of frequencies considered [1-10 Hz] and [ 4.5- 10 Hz], respectively.

- Line 275. Please specify this reference. See Line 48.

- Line 277. Are you sure that including $\Pi$ in eq. 2 is correct? Are you considering radians?

- Lines 288 - Specify in the Conclusions that the results correspond to the case of study, for a powder-snow avalanche recorded at 1000 m from the starting point.

- Line 292. ....source mechanism of the infrasound

- Line 293. Specify the wave parameters (back-azimuth and apparent velocity) or rephrase the two sentences.

- Line 294....purposes in the case that a powder part develops.

- Line 295. What happens if there is a sharp change in the slope were a powder part is also developed? See e.g. https://www.youtube.com/watch?v=WAbIcWxwGg4

- Line 303. Please indicate in % what it means strongly affected. In addition, you must consider the different frequency content of the two time series in your calculations.

- Line 313. Energy radiation

- References Please, Indicate correctly the spelling of the surnames.

-Figures - Figure 1. In Figure 1b) the s7 sensor is missed.

Figure Caption 1. Replace "array" by (c) arrays. Indicate the meaning of si and mi.

- Figure Caption 2. Replace runout zone by maximum runout zone.

- Figure Caption 3 Specify the array (infrasound?). The arrays are distant 500 m and the scale is not included.

- Figure 4. Redraw figures c) and d) according to my previous comments

- Figure 5c) Replace spectal by spectral.

- Figure 6. In Figure 6b) convert counts to ground speed and include in the horizontal axis the title like Figure 5a. For the benefit of the comparison, change the vertical scale on a more detailed scale for the posterior azimuth 6d) and the apparent velocity 6f) of the seismic data, even if you lose some outliers.

- Figure 7. Indicate units, when necessary, in the Figure and in the Figure caption. °N is it correct in Figure 7c).

- Figure Caption 8. Indicate the location of the arrays.

Please also note the supplement to this comment:
https://www.earth-surf-dynam-discuss.net/esurf-2019-61/esurf-2019-61-RC1-
supplement.pdf

---

## Referee Comment (RC2) · Bradley Lipovsky (Referee) · 8 Jan 2020

To Whom It May Concern:

The authors compare infrasonic and seismic observations of a dry snow avalanche. This work constitutes a substantial contribution to scientific progress and deserves to be published in ESD.

I would suggest caution when considering the partitioning of radiated energy between infrasonic and seismic waves for several reasons. The results are somewhat limited by the use of a geophone which is relatively insensitive to low frequencies. I furthermore would generally expect the seismic signal to be lower frequency than the infrasound. The simple equation used in Equation 1 does not account for the frequency depen-

dence of waves through a porous snow layer and should be taken with a grain of salt. It also does not account for the specific generation of surface waves which the authors later claim to be important.

I would generally recommend clarifying the distinction between observations and inter-pretations/results/models. Examples include: First paragraph of Section 4 talks about seismo-acoustic records and their interpretation at the same time. Section 3 (line 104 on) talks about the model. Section 5 largely consists of discussion points. It would improve the readability of the paper to follow a more traditional structure; i.e., Data, Methods, Results, Discussion.

Figure 6 is in units of counts rather than m/s, which makes it difficult for the reader to asses the scale.

Line 92-93 Two seconds error seems like rather poor timing. Did any of the instruments use GPS for timing?

Could the observations be related to recent work suggesting a more nuanced avalanche classification system (i.e., Kohler et al., 2018 10.1002/2017JF004375)?

I applaud the authors for putting their data in an Open Science Framework Repository.

Sincerely,

Brad Lipovsky

---

## Author Comment (AC1) · 28 Jan 2020

On behalf of my co-authors, Alec van Herwjinen, Marc Christen, Maria Cristina Silengo and Giulia Barfucci, I would like to thank Emma Surinach of her careful and objective review that allowed us to strongly improve the manuscript. Emma pointed out clearly the most critical points. Following her comments, in particular, we discussed in much better detail teh work performed by Kogelnig et al., 2011, that is a very important paper in the field of experimental geophysical analysis of snow avalanches, and highlighted common findings as well as discrepancies. In general our finding are in very good agreement with the ones by Kogelnig et al., 2011, despite a different approach and sensor configuration is used. We also highlighted clearly the limit we have in this manuscript concerning the frequency response of the seismic array (>4.5 Hz) and

agree with teh reviewer that for future deployment we will definitely deploy sensors with a broader frequency response. All comments raised by the reviewer were addressed in the text, and when required new figures where realized. In general we are very happy with the comments of the reviewer and the revised version of the manuscript. Below you will find a point-by-point reply to the comments of teh reviewer. Sincerely, Emanuele Marchetti

Reviewer: It is a very interesting contribution, although the authors have not take into account the previous paper of Kogelnig et al., (2011) where seismic and infrasound time series are compared for four different types of avalanches at the VdLS experimental site, descending along different paths, unlike the one presented in the manuscript under review in which only one avalanche is studied. In this paper the infrasound and seismic time series obtained in collocated sensors with a common time are compared, and also with the time series obtained in two more seismometers placed along the avalanche path. Additionally, a comparison was made with other "in situ" direct measurements (flow depth and internal velocities). The frequencies involved in the study are in the range of [1-40] Hz for both type of measurements. Of interest is the content of "low" [1-3] Hz frequencies of the seismic when comparing with the infrasound. Because of the completeness of these data, with respect to that of the data of the manuscript under review, the authors must take into account in their discussion and conclusions the results obtained previously. In principle, part of the obtained results by the authors could confirm the previous ones or contradict them. Reply: We are perfectly aware of the work performed by Kogeling et al., 2011, and it was already referenced in the text. However, following the comment of the reviewer, we included a better discussion of our findings compared to what presented already by Kogeling et al., 2011.

Reviewer: The use of the combination of the two arrays in this study is very positive, but the authors must be aware of the limitations of their study. In addition, the results presented also depend on the specific topography. One of the difficulties of the comparison of the results of the two arrays is that the infrasound array has a direct view of

the avalanche flow and the array of geophones does not. What would happen in the case of the existence of a shadow zone for the infrasound? Or if the seismic array had been collocated with the infrasound array? Reply: Infrasound propagation is strongly affected by the local topography. Basically, moving sources that are not line-of-sight to the infrasound array, are recorded with a limited variation of back/azimuth and/or apparent velocity. This is not the case of seismic data, that propagate in the ground, even-if, in this case, variable phase can make the analysis more complicated. This aspect has been further discussion in the manuscript. In case of a collocated seismic and infrasound array, p-wave induced variation of back-azimuth would mimic the back-azimuth variation induced by infrasound.

Reviewer: As regards Section 5. Kogelnig et al. (2011) includes a section dedicated to the source of infrasound and seismic signals. There, a synthetic signal is obtained using the ex- pression of Ffowcs Williams (1963) that describes the acoustic intensity generated by a turbulent source in motion. The modeling results are compared with the infrasound time series obtained from an avalanche. In addition, due to the existence of a suspension layer that can generate infrasound, an explanation is included for not considering a unic specific source of infrasound. Reply: Following the comment of the reviewer, we included a comment this aspect in the discussion of the source mechanism of the infrasound signal.

Reviewer: In addition, the flow dimension D is calculated for the dominant frequencies. The authors must take into account in their discussion and conclusions the results obtained previously. Reply: We completely agree with the reviewer. A discussion about this aspect, following the work performed by Kogeling et al., 2011, is included in the manuscript.

Reviewer: A remark on Figure 4. This figure is very important in the interpretation of the time series and the results. Note that the origin of the time series corresponds to the farthest distances. To facilitate interpretation with the time series, the authors must convert the distances into time (using the obtained speeds) and reverse the origin of the

distance. In addition, the slope angle (derived from the profile) incorporated in Figure 4d with the outputs of the RAMMS model will help to better correlate the slope change with the features of the time series. Reply: We thank the reviewer for this comment and we changed the figure. In particular we reversed the distance and divided the flow depth and flow velocity in two different subplots, and over-imposed that in the path profile, in order to highlight correlation of avalanche parameters and path geometry. Following the comment of the reviewer, we also tried converting distance into time, but found the output figure misleading and of difficult comprehension at this stage of the manuscript, where the recorded infrasound and seismic time series are not introduced yet. The new figure, realized after some of the comments of the reviewer, is definitely much clearer. The figure caption has been chaged accordingly.

Reviewer: The limitation in the frequency content used in the study must be indicated in the abstract. Reply: This is now included in the abstract.

Reviewer: - Line 48. Please, confirm that reference Naugolnykh, K., and Bedard, A is correct. There are different contributions of these authors with the same title, e.g. Reply: The reference has been double-checked and corrected. The correct reference is: Naugolnykh, K., and Bedard, A., (2002): A model of the avalanche infrasonic radiation. A IEEE International Geoscience and Remote Sensing Symposium DOI: 10.1109/IGARSS.2002.1025713.

Reviewer: Lines 76- 90. -Line 77. Please, check the figure numbers. e.g. Is figure 2c correct or it is 1b? Reply: The reviewer is perfectly right. This was corrected in the text.

Reviewer: Although it is indicated in the abstract you must mention here the distance between the arrays -Line 85. Figure 1c? Reply: This information has been added in the text.

Reviewer: -Lines 91-93. This assertion will be correct assuming that the earthquake is recorded in the infrasound sensors. An explanation on this, references, or more detail is needed. ... Reply: Seismic ground shaking is routinely recorded

on infrasound sensors. The shaking of the ground causes a movement of the infrasound sensor and therefore a variation of pressure. Just to provide an example, the datasheet of mb3 sensor by seismowave provide the seismic sensitivity (http://seismowave.com/medias/documents/MB3a.pdf). We believe that a reference here is probably redundant. We added a reference of an infrasound study of an earthquake in Italy (Marchetti et al., SRL, 2016), where the seismic shaking is recorded by the microphone well below the secondary infrasound produced by ground shaking at the epicenter.

Reviewer: <2 s assuming the difference in wave travel time and wave propagation speeds of ... and a distance of ... -Line 100. Please, check if Figure 3 is correct. Reply: Following the comment of the reviewer, more detail is provided in the text.

Reviewer: - Line 123. Indicate which sensor corresponds to the time series presented in Figures 5 a and b. Or are they stacked time series? Reply: The time series in Figure 5 a and b correspond to the 3rd sensor of both arrays. We specified that in the figure caption.

Reviewer: - Line 131. Are you sure about 35 s? Could you specify the signal limits here even if you do it below? Reply: This value is not correct. The infrasound signal is approximately 45 seconds. We corrected the text.

Reviewer: - Lines 140-143. In fact, there are two speeds, one of the infrasound waves in the air and the other corresponding to the source (avalanche). Authors should specify this somewhere, here or above in the presentation of the method. Reply: In the work we never deal with the velocity of the avalanche.

Reviewer: - Line 147 February Reply: Corrected in the text.

Reviewer: - Line 149. Note that the only effect of the low pass filter is in the infrasound, because the geophones natural frequency is 4.5 Hz Reply: Yes, the reviewer is perfectly right, thanks for the note.
Reviewer: - Line 160 approx. 35 s as indicated in Line 131. Reply: This was fixed in the text.

Reviewer: - Line 162. An explanation of the difference between the detection of time arrival of the matrix and that of the seismic amplitudes observed at 18.30 (Figure 5a) that are clearly due to the avalanche is necessary. Given that the velocity of the seismic waves in relation to the avalanche speed and that of the infrasound in the air, it seems that the avalanche started earlier than indicated. Reply: Around 18:30 we obtain the first clear detection of seismic signal produced by the avalanche. The first detection of infrasound is recorded after, both because it started to be produced after during the flow and because propagation speed of infrasound is lower.

Reviewer: - Lines 167-169. Does it refer to the processing of the seismic array or that of in- frasound? Please specify. Vilajosana et al. (2007a) obtained the mentioned ground phase velocities from waves generated by explosions at Ryggfonn. These speeds are independent of avalanches. This is a feature of the site. I think there is a misunderstanding. Please clarify. Reply: Line 167-169 refers to the processing of seismic wave. We clarified that in the text.

Reviewer: - Line 176. Section 5. See previous comment on this. Reply: DONE Reviewer: - Line 181. Auxiliary material. Do you mean the video? Include the reference. Reply: Yes, this is what we mean. Text was corrected accordingly.

Reviewer: - Line 181. ...radiated from a point source Reply: Corrected in the text.

Reviewer: - Line 184. ... along the path considering a point source Reply:Corrected in the text.

Reviewer: - Lines 210-211. This could be an effect of the relative position of the arrays as men- tioned by the authors in Line 214. Reply: This is a possibility and is discussed in the text.

Reviewer: - Line 227. With your results, there is not enough information to generalize

to all the avalanches, in plural. Reply: We are quite confident that snow avalanches are characterized by a predominant source of infrasound energy. Based on previous experimental studies this is likely the front. This is very different for example from other density currents such as lahars and debris flows, where array processing identifies a clear lack of coherence that is interpreted as muktiple sources acting at once. This aspect has been clarified in the text.

Reviewer: - Line 232. Remember the content in Kogelnig et al., (2011). Reply: The study of Kogelnig is discussed in detail in the revised manuscript. They analyse infrasound in terms of the elastic energy radiated by the turbulent flow at the avalanche front. Results are very good. In terms of array analysis however, it can be considerd as a point source moving downhill. Therefore we believe that additional comment here, once the point above has been addressed are redundant. Moreover, in Kogeling too a point source is assumed to estimate the size (D) that is eventually used to calculate infrasound energy radiation by the turbulent flow.

Reviewer: - Line 246. and references therein.... Reply: Corrected in the text

Reviewer: - Line 249. Note that the effect of filtering from 1 to 10 Hz and realistically, from 4.5 to 10 Hz, also presents problems in the quantification of the energy, since part of the signal is lost. Reply: Following the suggestion of the reviewer, the frequency limitation of our study has been highlighted clearly in the text.

Reviewer: - Line 252. This was also mentioned in Kogelnig et al., (2011) Reply: The reference is added in the text.

Reviewer: - Line 260-264. The authors must specify that this is in the range of frequencies con- sidered [1-10 Hz] and [ 4.5- 10 Hz], respectively. Reply: Same as above. This has been stated clearly in the abstract, methodology and conclusions.

Reviewer: - Line 275. Please specify this reference. See Line 48. Reply: Reference has been corrected in the text and reference list.

Reviewer: - Line 277. Are you sure that including $\Pi$ in eq. 2 is correct? Are you considering radians? Reply: Yes. Considering a regid sphere that starts moving into the atmosphere, the characteristic angular frequency is proportional to the ratio between the sound wave propagation velocity and the radius of the sphere, resulting into equation 2. This equation, proposed to investigate the frequency of infrasound radiated by a snow avalanche by Naugolnykh, K., and Bedard, A., was developed already by Landau and Liftshitz, (Fluid Mechanics, 1959, page 287). We prefer referencing to the work of Naugolnykh, K., and Bedard, A., that applied the physics to snow avalanches, rather than a basic book of Fluid Mechanics.

Reviewer: - Lines 288 - Specify in the Conclusions that the results correspond to the case of study, for a powder-snow avalanche recorded at 1000 m from the starting point. Reply: This has been specified in the text.

Reviewer: - Line 292. ....source mechanism of the infrasound Reply: Corrected in the text.

Reviewer: - Line 293. Specify the wave parameters (back-azimuth and apparent velocity) or rephrase the two sentences. Reply: Corrected in the text.

Reviewer: - Line 294....purposes in the case that a powder part develops. Reply:Corrected in the text.

Reviewer: - Line 295. What happens if there is a sharp change in the slope were a powder part is also developed? See e.g. https://www.youtube.com/watch?v=WAbIcWxwGg4 Reply: According to our findings and experience, infrasound is strongly controlled by the evolution of the powder cloud. Therefore, stable infrasound will be produced at the site where slope changes. Such a behavior was observed also for debris flow (Marchetti et al., 2019, JGR) ad enhanced infrasound is radiated at waterfalls that are detected despite the flow keeps moving downhill.

**ESurfD**
[Figure]

Reviewer: - Line 303. Please indicate in % what it means strongly affected. In addition, you must consider the different frequency content of the two time series in your calculations. Reply: This has been addressed in the text.

Reviewer: - Line 313. Energy radiation Reply: This has been corrected in the text.

Reviewer: - References Please, Indicate correctly the spelling of the surnames. Reply: Reference list has been doublechecked carefully.

Reviewer: -Figures - Figure 1. In Figure 1b) the s7 sensor is missed. Reply: There was s1 twice. This was corrected in the figure.

Reviewer: Figure Caption 1. Replace "array" by (c) arrays. Indicate the meaning of si and mi. Reply: Figure caption was corrected. s1-s7 are seismometers 1-7 and m1-m5 are microphones 1-5.We think that specifying that is probably redundant.

Reviewer: - Figure Caption 2. Replace runout zone by maximum runout zone. Reply: Figure caption has been corrected.

Reviewer: - Figure Caption 3 Specify the array (infrasound?). The arrays are distant 500 m and the scale is not included. Reply: Figure caption has been corrected.

Reviewer: - Figure 4. Redraw figures c) and d) according to my previous comments Reply: Subplots c and d have been replaced.

Reviewer: - Figure 5c) Replace spectal by spectral. Reply: Figure has been corrected.

Reviewer: - Figure 6. In Figure 6b) convert counts to ground speed and include in the horizontal axis the title like Figure 5a. For the benefit of the comparison, change the vertical scale on a more detailed scale for the posterior azimuth 6d) and the apparent velocity 6f) of the seismic data, even if you lose some outliers. Reply: Figure 6 has been modified following all the suggestions of the reviewer.

Reviewer: - Figure 7. Indicate units, when necessary, in the Figure and in the Figure caption. âŮęN is it correct in Figure 7c). Reply: Figure 7 has been corrected according

to the comment of the reviewer.

Reviewer: - Figure Caption 8. Indicate the location of the arrays. C6 Reply: The figure caption has been corrected following the comment of the reviewer.

---

## Author Comment (AC2) · 28 Jan 2020

On behalf of my co-authors, I would like to tank Dr. Lipovsky for the careful and objective review. We addressed all the comments of the reviewer and this allowed, in our opinion, to greatly improve the work. In particular we discussed in more detail the energy partitioning, as well as equation 1, by taking into account the limited frequency response of the geophones. Moreover, we followed the suggestion of the reviewer and reorganized the manuscript in a more standard form, and this lead to a better readability of the text and clarity of the content. Below you will find a point-by-point reply to the comments of the reviewer. Sincerely, Emanuele Marchetti

\_\_\_\_\_\_\_\_\_\_\_\_\_\_\_\_\_\_\_\_\_\_\_\_\_\_\_\_\_\_\_\_\_\_\_\_\_\_\_\_\_\_\_\_\_\_\_\_\_\_\_\_\_\_\_\_\_\_

[Figure]

Reviewer: I would suggest caution when considering the partitioning of radiated energy between infrasonic and seismic waves for several reasons. The results are somewhat limited by the use of a geophone which is relatively insensitive to low frequencies. I furthermore would generally expect the seismic signal to be lower frequency than the infrasound. Reply: Unfortunately, The high frequency response of the geophones is a clear limitation. This was correctly pointed out from both reviewers. Therefore, we pointed out such a limitation in the text, wherever it applies, in order to make clear that the results might be partly affected by the geophone frequency response. Definitely, in the future we will have to deploy a broadband seismometer collocated with the infrasound array.

Reviewer: The simple Equation 1 does not account for the frequency dependence of waves through a porous snow layer and should be taken with a grain of salt. It also does not account for the specific generation of surface waves which the authors later claim to be important. Reply: Equation 1 describes the transition of infrasound wave (longitudinal waves) to the ground as vertical seismic velocity (body waves). It depends solely from the elastic constant of the medium ($\mu$ and l). This aspect has been clarified in the text.

Reviewer: I would generally recommend clarifying the distinction between observations and interpretations/results/models. Examples include: First paragraph of Section 4 talks about seismo-acoustic records and their interpretation at the same time. Section 3 (line 104 on) talks about the model. Section 5 largely consists of discussion points. It would improve the readability of the paper to follow a more traditional structure; i.e., Data, Methods, Results, Discussion. Reply: The paper was re-organized following the suggestion of the reviewer.

Reviewer: Figure 6 is in units of counts rather than m/s, which makes it difficult for the reader to asses the scale. Reply: We changed the scale of the figure into m/s.

Reviewer: Line 92-93 Two seconds error seems like rather poor timing. Did any of

the instruments use GPS for timing? Reply: The infrasound array is equipped with a GPS receiver for time synchronization (line 89). The seismic array was not, and used a GPS receiver to synchronize the acquisition computer clock. There might be an error here. Therefore, we aligned seismic and infrasound data with the occurrence of local earthquakes that were recorded by both system. Here comes the error of two seconds, that despite being very large considering GPS timing, is very low for the aim of our research, that aims to compare seismic and infrasound data at the timescale of the event (tens of seconds).

Reviewer: Could the observations be related to recent work suggesting a more nuanced avalanche classification system (i.e., Kohler et al., 2018 10.1002/2017JF004375)? Reply: A reference to the work by Kohler et al., 2018 has been included in the introduction when describing the PSA that basically corresponds to the Intermediate Regime identified by Kohler et al., 2018.

---

## Author Response (AR1)

Emanuele Marchetti
Department of Earth Sciences,
University of Firenze
Via G. La Pira, 4
50121, Firenze
Italy

To the Editor of the
Earth Surface Dynamics

Firenze, February 24, 2020

Dear Editor,

I submit to your attention for publication on Earth Surface Dynamics the manuscript "Seismo-acoustic energy partitioning of a powder snow avalanche", written in cooperation and agreement with Alec van Herwijnen, Marc Christen, Maria Cristina Silengo and Giula Barfucci. The manuscript was revised following the comments of the two reviewers, Emma Surinach and Bradley Lipovsky.

All the comments of the reviewers have been addressed in the text and we are grateful as they allowed us to greatly improve the manuscript. Below we provide a point-by-point reply to the comments of the reviewers and a version of the manuscript with track changes.

Hoping in your interest in this topic, I thank you in advance for your time.

Sincerely
Emanuele Marchetti

[Figure]

**Reply to the review by Bradley Lipovsky**

*Reviewer: I would suggest caution when considering the partitioning of radiated energy between infrasonic and seismic waves for several reasons. The results are somewhat limited by the use of a geophone which is relatively insensitive to low frequencies. I furthermore would generally expect the seismic signal to be lower frequency than the infrasound.*

Unfortunately, the high frequency response of the geophone is a clear limitation. This was correctly pointed out from both reviewers. Therefore, we pointed out such a limitation in the text, wherever it applies, in order to make clear that the results might be partly affected by the geophone frequency response. Definitely, in the future we will have to deploy a broadband seismometer collocated with the infrasound array.

*Reviewer: The simple Equation 1 does not account for the frequency dependence of waves through a porous snow layer and should be taken with a grain of salt. It also does not account for the specific generation of surface waves which the authors later claim to be important.*

Equation 1 describes the transition of infrasound wave (longitudinal waves) to the ground as vertical seismic velocity (body waves). It depends solely from the elastic constant of the medium ($\mu$ and l). This aspect has been clarified in the text.

*Reviewer: I would generally recommend clarifying the distinction between observations and interpretations/results/models. Examples include: First paragraph of Section 4 talks about seismo-acoustic records and their interpretation at the same time. Section 3 (line 104 on) talks about the model. Section 5 largely consists of discussion points. It would improve the readability of the paper to follow a more traditional structure; i.e., Data, Methods, Results, Discussion.*

The paper was re-organized following the suggestion of the reviewer. All the changes are clearly highlighted in the track changes version.

*Reviewer: Figure 6 is in units of counts rather than m/s, which makes it difficult for the reader to asses the scale.*

We changed the scale of the figure into m/s.

*Reviewer: Line 92-93 Two seconds error seems like rather poor timing. Did any of the instruments use GPS for timing?*

The infrasound array is equipped with a GPS receiver for time synchronization (line 89). The seismic array was not, and used a GPS receiver to synchronize the acquisition computer clock. There might be an error here. Therefore, we aligned seismic and infrasound data with the occurrence of local earthquakes that were recorded by both system. Here comes the error of two seconds, that despite being very large considering GPS timing, is very low for the aim of our research, that aims to compare seismic and infrasound data at the timescale of the event (tens of seconds).

[Figure]

*Reviewer: Could the observations be related to recent work suggesting a more nuanced avalanche classification system (i.e., Kohler et al., 2018 10.1002/2017JF004375)?*

A reference to the work by Kohler et al., 2018 has been included in the introduction when describing the PSA that basically corresponds to the Intermediate Regime identified by Kohler et al., 2018.

[Figure]

**Reply to the review by Emma Surinach**

*Reviewer: It is a very interesting contribution, although the authors have not take into account the previous paper of Kogelnig et al., (2011) where seismic and infrasound time series are compared for four different types of avalanches at the VdLS experimental site, descending along different paths, unlike the one presented in the manuscript under review in which only one avalanche is studied. In this paper the infrasound and seismic time series obtained in collocated sensors with a common time are compared, and also with the time series obtained in two more seismometers placed along the avalanche path. Additionally, a comparison was made with other "in situ" direct measurements (flow depth and internal velocities). The frequencies involved in the study are in the range of [1-40] Hz for both type of measurements. Of interest is the content of "low" [1-3] Hz frequencies of the seismic when comparing with the infrasound. Because of the completeness of these data, with respect to that of the data of the manuscript under review, the authors must take into account in their discussion and conclusions the results obtained previously. In principle, part of the obtained results by the authors could confirm the previous ones or contradict them.*

We are perfectly aware of the work performed by Kogelnig et al., 2011, and it was already referenced in the text. However, following the comment of the reviewer, we included a better discussion of our findings compared to what presented already by Kogelnig et al., 2011.

*Reviewer: The use of the combination of the two arrays in this study is very positive, but the authors must be aware of the limitations of their study. In addition, the results presented also depend on the specific topography. One of the difficulties of the comparison of the results of the two arrays is that the infrasound array has a direct view of the avalanche flow and the array of geophones does not. What would happen in the case of the existence of a shadow zone for the infrasound? Or if the seismic array had been collocated with the infrasound array?*

Infrasound propagation is strongly affected by the local topography. Basically, moving sources that are not line-of-sight to the infrasound array, are recorded with a limited variation of back/azimuth and/or apparent velocity. This is not the case of seismic data, that propagate in the ground, even if, in this case, variable seismic phases can make the analysis more complicated. This aspect has been further discussion in the manuscript. In case of a collocated seismic and infrasound array, p-wave induced variation of back-azimuth would mimic the back-azimuth variation induced by infrasound.

*Reviewer: As regards Section 5. Kogelnig et al. (2011) includes a section dedicated to the source of infrasound and seismic signals. There, a synthetic signal is obtained using the expression of Ffowcs Williams (1963) that describes the acoustic intensity generated by a turbulent source in motion. The modeling results are compared with the infrasound time series obtained from an avalanche. In addition, due to the existence of a suspension layer that can generate infrasound, an explanation is included for not considering a unic specific source of infrasound.*

[Figure]

Following the comment of the reviewer, we included a comment this aspect in the discussion of the source mechanism of the infrasound signal.

*Reviewer: In addition, the flow dimension D is calculated for the dominant frequencies. The authors must take into account in their discussion and conclusions the results obtained previously.*

We completely agree with the reviewer. A discussion about this aspect, following the work performed by Kogeling et al., 2011, is included in the manuscript.

*Reviewer: A remark on Figure 4. This figure is very important in the interpretation of the time series and the results. Note that the origin of the time series corresponds to the farthest distances. To facilitate interpretation with the time series, the authors must convert the distances into time (using the obtained speeds) and reverse the origin of the distance. In addition, the slope angle (derived from the profile) incorporated in Figure 4d with the outputs of the RAMMS model will help to better correlate the slope change with the features of the time series.*

We thank the reviewer for this comment and we changed the figure. In particular we reversed the distance and divided the flow depth and flow velocity in two different subplots, and over-imposed that in the path profile, in order to highlight correlation of avalanche parameters and path geometry. Following the comment of the reviewer, we also tried converting distance into time, but found the output figure misleading and of difficult comprehension at this stage of the manuscript, where the recorded infrasound and seismic time series are not introduced yet. The new figure, realized after some of the comments of the reviewer, is definitely much clearer. The figure caption has been chaged accordingly.

*Reviewer: The limitation in the frequency content used in the study must be indicated in the abstract.*

This is now included in the abstract.

*Reviewer: - Line 48. Please, confirm that reference Naugolnykh, K., and Bedard, A is correct. There are different contributions of these authors with the same title, e.g.*

The reference has been double-checked and corrected. The correct reference is:

Naugolnykh, K., and Bedard, A., (2002): A model of the avalanche infrasonic radiation. A IEEE International Geoscience and Remote Sensing Symposium DOI: 10.1109/IGARSS.2002.1025713.

*Reviewer: Lines 76- 90. -Line 77. Please, check the figure numbers. e.g. Is figure 2c correct or it is 1b?*

The reviewer is perfectly right. This was corrected in the text.

*Reviewer: Although it is indicated in the abstract you must mention here the distance between the arrays -Line 85. Figure 1c?*

[Figure]

This information has been added in the text.

*Reviewer: -Lines 91-93. This assertion will be correct assuming that the earthquake is recorded in the infrasound sensors. An explanation on this, references, or more detail is needed. ...*

Seismic ground shaking is routinely recorded on infrasound sensors. The shaking of the ground causes a movement of the infrasound sensor and therefore a variation of pressure. Just to provide an example, the datasheet of mb3 sensor by seismowave provide the seismic sensitivity (http://seismowave.com/medias/documents/MB3a.pdf). We believe that a reference here is probably redundant.

*Reviewer: <2 s assuming the difference in wave travel time and wave propagation speeds of ... and a distance of ... -Line 100. Please, check if Figure 3 is correct.*

Following the comment of the reviewer, more detail is provided in the text.

*Reviewer: - Line 123. Indicate which sensor corresponds to the time series presented in Figures 5 a and b. Or are they stacked time series?*

The time series in Figure 5 a and b correspond to the 3$^{rd}$ sensor of both arrays. We specified that in the figure caption.

*Reviewer: - Line 131. Are you sure about 35 s? Could you specify the signal limits here even if you do it below?*

This value is not correct. The infrasound signal is approximately 45 seconds. We corrected the text.

*Reviewer: - Lines 140-143. In fact, there are two speeds, one of the infrasound waves in the air and the other corresponding to the source (avalanche). Authors should specify this somewhere, here or above in the presentation of the method.*

In the work we never deal with the velocity of the avalanche.

*Reviewer: - Line 147 February*

Corrected in the text.

*Reviewer: - Line 149. Note that the only effect of the low pass filter is in the infrasound, because the geophones natural frequency is 4.5 Hz*

Yes, the reviewer is perfectly right, thanks for the note.

*Reviewer: - Line 160 approx. 35 s as indicated in Line 131.*

This was fixed in the text.

[Figure]

*Reviewer: - Line 162. An explanation of the difference between the detection of time arrival of the matrix and that of the seismic amplitudes observed at 18.30 (Figure 5a) that are clearly due to the avalanche is necessary. Given that the velocity of the seismic waves in relation to the avalanche speed and that of the infrasound in the air, it seems that the avalanche started earlier than indicated.*

Around 18:30 we obtain the first clear detection of seismic signal produced by the avalanche. The first detection of infrasound is recorded after, both because it started to be produced after during the flow and because propagation speed of infrasound is lower.

*Reviewer: - Lines 167-169. Does it refer to the processing of the seismic array or that of in- frasound? Please specify. Vilajosana et al. (2007a) obtained the mentioned ground phase velocities from waves generated by explosions at Ryggfonn. These speeds are independent of avalanches. This is a feature of the site. I think there is a misunder- standing. Please clarify.*

Line 167-169 refers to the processing of seismic wave. We clarified that in the text.

*Reviewer: - Line 176. Section 5. See previous comment on this.*

*DONE*

*Reviewer: - Line 181. Auxiliary material. Do you mean the video? Include the reference.*

Yes, this is what we mean. Text was corrected accordingly.

*Reviewer: - Line 181. ..radiated from a point source*

Corrected in the text.

*Reviewer: - Line 184. ... along the path considering a point source*

Corrected in the text.

*Reviewer: - Lines 210-211. This could be an effect of the relative position of the arrays as men- tioned by the authors in Line 214.*

This is a possibility and is discussed in the text.

*Reviewer: - Line 227. With your results, there is not enough information to generalize to all the avalanches, in plural.*

We are quite confident that snow avalanches are characterized by a predominant source of infrasound energy. Based on previous experimental studies this is likely the front. This is very different for example from other density currents such as lahars and debris flows, where array processing identifies a clear lack of coherence that is interpreted as muktiple sources acting at once. This aspect has been clarified in the text.

[Figure]

*Reviewer: - Line 232. Remember the content in Kogelnig et al., (2011).*

The study of Kogelnig is discussed in detail. They analyse infrasound in terms of the elastic energy radiated by the turbulent flow at the avalanche front. Results are very promising. In terms of array analysis however, it can be considerd as a point source moving downhill. Therefore we believe that additional comment here, once the point above has been addressed are redundant. Moreover, in Kogeling too a point source is assumed to estimate the size (D) that is eventually used to calculate infrasound energy radiation by the turbulent flow.

*Reviewer: - Line 246. and references therein....*

Corrected in the text

*Reviewer: - Line 249. Note that the effect of filtering from 1 to 10 Hz and realistically, from 4.5 to 10 Hz, also presents problems in the quantification of the energy, since part of the signal is lost.*

Following the suggestion of the reviewer, the frequency limitation of our study has been highlighted clearly in the text.

*Reviewer: - Line 252. This was also mentioned in Kogelnig et al., (2011)*

The reference is added in the text.

*Reviewer: - Line 260-264. The authors must specify that this is in the range of frequencies con- sidered [1-10 Hz] and [ 4.5- 10 Hz], respectively.*

Same as above. This has been stated clearly in the abstract, methodology and conclusions.

*Reviewer: - Line 275. Please specify this reference. See Line 48.*

Reference has been corrected in the text and reference list.

*Reviewer: - Line 277. Are you sure that including Π in eq. 2 is correct? Are you considering radians?*

Yes. Considering a regid sphere that starts moving into the atmosphere, the characteristic angular frequency is proportional to the ratio between the sound wave propagation velocity and the radius of the sphere, resulting into equation 2. This equation, proposed to investigate the frequency of infrasound radiated by a snow avalanche by Naugolnykh, K., and Bedard, A., was developed already by Landau and Liftshitz, (Fluid Mechanics, 1959, page 287). We prefer referencing to the work of Naugolnykh, K., and Bedard, A., that applied the physics to snow avalanches, rather than a basic book of Fluid Mechanics.

*Reviewer: - Lines 288 - Specify in the Conclusions that the results correspond to the case of study, for a powder-snow avalanche recorded at 1000 m from the starting point.*

[Figure]

This has been specified in the text.

*Reviewer: - Line 292. ....source mechanism of the infrasound*

Corrected in the text.

*Reviewer: - Line 293. Specify the wave parameters (back-azimuth and apparent velocity) or rephrase the two sentences.*

Corrected in the text.

*Reviewer: - Line 294....purposes in the case that a powder part develops.*

Corrected in the text.

*Reviewer: - Line 295. What happens if there is a sharp change in the slope were a powder part is also developed? See e.g. https://www.youtube.com/watch?v=WAbIcWxwGg4*

According to our findings and experience, infrasound is strongly controlled by the evolution of the powder cloud. Therefore, stable infrasound will be produced at the site where slope changes. Such a behavior was observed also for debris flow (Marchetti et al., 2019, JGR) ad enhanced infrasound is radiated at waterfalls that are detected despite the flow keeps moving downhill.

*Reviewer: - Line 303. Please indicate in % what it means strongly affected. In addition, you must consider the different frequency content of the two time series in your calculations.*

This has been addressed in the text.

*Reviewer: - Line 313. Energy radiation*

This has been corrected in the text.

*Reviewer: - References Please, Indicate correctly the spelling of the surnames.*

Reference list has been doublechecked carefully.

*Reviewer: -Figures - Figure 1. In Figure 1b) the s7 sensor is missed.*

There was s1 twice. This was corrected in the figure.

*Reviewer: Figure Caption 1. Replace "array" by (c) arrays. Indicate the meaning of si and mi.*

Figure caption was corrected. s1-s7 are seismometers 1-7 and m1-m5 are microphones 1-5.We think that specifying that is probably redundant.

[Figure]

*Reviewer: - Figure Caption 2. Replace runout zone by maximum runout zone.*

Figure caption has been corrected.

*Reviewer: - Figure Caption 3 Specify the array (infrasound?). The arrays are distant 500 m and the scale is not included.*

Figure caption has been corrected.

*Reviewer: - Figure 4. Redraw figures c) and d) according to my previous comments*

Subplots c and d have been replaced.

*Reviewer: - Figure 5c) Replace spectal by spectral.*

Figure has been corrected.

*Reviewer: - Figure 6. In Figure 6b) convert counts to ground speed and include in the horizontal axis the title like Figure 5a. For the benefit of the comparison, change the vertical scale on a more detailed scale for the posterior azimuth 6d) and the apparent velocity 6f) of the seismic data, even if you lose some outliers.*

Figure 6 has been modified following all the suggestions of the reviewer.

*Reviewer: - Figure 7. Indicate units, when necessary, in the Figure and in the Figure caption. °N is it correct in Figure 7c).*

Figure 7 has been corrected according to the comment of the reviewer.

*Reviewer: - Figure Caption 8. Indicate the location of the arrays. C6*

The figure caption has been corrected following the comment of the reviewer.

[revised manuscript text omitted]

---

## Author Response (AR2)

UNIVERSITÀ
DEGLI STUDI
FIRENZE
DST
DIPARTIMENTO DI
SCIENZE DELLA TERRA

Emanuele Marchetti
Department of Earth Sciences,
University of Firenze
Via G. La Pira, 4
50121, Firenze
Italy

To the Editor of the
Earth Surface Dynamics

Firenze, April 1, 2020

Dear Editor,

I submit to your attention for publication on Earth Surface Dynamics the manuscript "Seismo-acoustic energy partitioning of a powder snow avalanche", written in cooperation and agreement with Alec van Herwijnen, Marc Christen, Maria Cristina Silengo and Giula Barfucci. After the revision of the manuscript realized following the comments of the two reviewers, Emma Surinach and Bradley Lipovsky, this last version was corrected addressing the comments of the Associate Editor Claire Masteller.

***All the comments of the Associate Editor have been addressed in the text.***

Hoping that you will find all the corrections satisfactory, I thank you in advance for your time.

Sincerely
Emanuele Marchetti